# Does Environmental Policy Affect Green Total Factor Productivity? Quasi-Natural Experiment Based on China’s Air Pollution Control and Prevention Action Plan

**DOI:** 10.3390/ijerph18158216

**Published:** 2021-08-03

**Authors:** Tinghui Li, Jiehua Ma, Bin Mo

**Affiliations:** 1School of Economics and Statistics, Guangzhou University, Guangzhou 510006, China; lith@gzhu.edu.cn (T.L.); 2112064101@e.gzhu.edu.cn (J.M.); 2Institute of Finance, Guangzhou University, Guangzhou 510006, China

**Keywords:** APCP Action Plan, *GTFP*, DID model, the rationalization of industrial structure, the optimization of industrial structure

## Abstract

It is the scientific way to promote the transformation and optimization of an industrial structure to promote the improvement of its green total factor productivity (*GTFP*) by formulating environmental regulation policies. Based on the **GTFP** panel data of 30 provinces in China from 2004 to 2017, this paper takes the “Air Pollution Control and Prevention Action Plan” (APCP Action Plan) as the proxy dummy variable of environmental regulation, and uses the difference-in-differences (DID) model to explore the impact of the implementation of the APCP Action Plan on *GTFP*. In addition, by constructing the industrial structure optimization index, this paper analyzes how the APCP Action Plan policy affects *GTFP* through the transformation and optimization of industrial structure. The following basic conclusions are obtained: First, environmental regulation policies like the APCP Action Plan can improve *GTFP*. Second, the APCP Action Plan has regional heterogeneity in promoting *GTFP* in different regions. The policy only significantly affects the *GTFP* in the Pearl River Delta region in southern China. Third, the “quantity” and “quality” of the optimization of industrial structure will weaken the promoting effect of the APCP Action Plan on *GTFP*. In contrast, the rationalization of industrial structure will aggravate this promoting effect.

## 1. Introduction

### 1.1. Background and Research Motivation

With the rapid development of a country’s economy and the continuous promotion of industrialization and urbanization, the consumption of energy resources continues to rise, issues such as air pollution and the excessive emission of carbon dioxide affect the daily lives of human beings. Therefore, for the sake of people’s health as well as a sustainable economic development, countries have put forward corresponding environmental protection policies and relevant legal provisions in view of these environmental problems. Different environmental policies have a complex relationship with a country’s economic “green” development. Environmental policy is a means of environmental control, which pays more attention to the efficient management of pollutants to improve people’s living environment and make it more comfortable and pleasant, which may decrease economic development. Judging from the intensity of environmental supervision brought by environmental policies due to the imbalance of economic development and industrial structure [1], different levels of pollution are subject to diverse levels of regulation. It will also lead to regional industrial transfer and the change of air pollution [2,3], thus making subtle changes in the sustainable economic development among regions. 

The Porter hypothesis holds that environmental regulation will promote the sustainable development of the economy by enhancing the level of innovation and the efficiency of energy utilization. It is essential to specifically study the impact of environmental policies on the *GTFP*. Air pollution control has become the most important part of developing a green ecological civilization in various countries. From 2012 to 2013, air pollution broke out in many regions of China, and the haze became more and more serious. To this end, the Chinese government issued the “Air Pollution Control and Prevention Action Plan” (APCP Action Plan for short) in September 2013 to improve the air quality. As the national strategy of improving air quality and the green sustainable development guide chart, the APCP Action Plan has high research value. Based on this, a quasi-natural experiment based on the APCP Action Plan can hopefully reflect the relationship between environmental policies and green total factor productivity. 

Controlling the emissions of air pollutants in China’s provinces and cities is particularly important in the APCP Action Plan. In addition to $PM_{2.5}$ and $PM_{10}$, which are the focus of the policy, the plan has reduced emissions of a variety of pollutants by strengthening environmental pollution control in industrial enterprises. First of all, the use of inefficient coal-fired boilers should be prohibited, the construction of desulfurization and dust removal projects in key industries should be strengthened. The transformation and upgrading of desulfurization and dust removal facilities should be accelerated in industrial enterprises. 

Secondly, the plan suggests the full implementation of a cleaner production model, cleaner production audit of key industries. Given the key areas and weak links of energy conservation and emission reduction, enterprises should adopt advanced technologies, processes and implement the technological transformation of cleaner production. Finally, all regions should control the total amount of coal consumption, speed up the alternative use of clean energy, promote the clean use of coal, improve the efficiency of energy use, and formulate enterprise emission quotas for wastewater, waste gas, and other pollutants by industries and regions, to strictly control the pollutant emission of enterprises. It can be seen that the APCP Action Plan not only controls $PM_{2.5}$ emissions, but also affects the emissions of industrial wastewater, fixed waste generation, sulfur dioxide and carbon dioxide. Therefore, this paper selects industrial wastewater, waste, sulfur dioxide, and carbon dioxide as the unexpected output in the calculation of *GTFP*, and because $PM_{2.5}$ is highly correlated with sulfur dioxide and other pollutants. The unexpected production does not consider the emission of $PM_{2.5}$. 

In sum, by exploring the relationship between the APCP Action Plan and *GTFP*, this paper further analyzes whether environmental policies can affect economic benefits and achieve sustainable development of the green economy while improving air quality and environmental protection levels.

### 1.2. Literature Review and Contribution

Existing related literature mainly studies the relationship between environmental regulation and regional productivity or environmental policy and regional sustainable development, mainly covering the following four aspects.

One is to study the impact of environmental policies on regional productivity. By studying the relationship between environmental policy and productivity in 18 OECD countries, Santis et al. [4] concluded that ecological policy would promote productivity growth through innovative development. Taking China’s sulfur dioxide emission trading pilot project as the quasi-natural experiment, Peng et al. [5] used panel data of Chinese industrial enterprises from 1998 to 2007 and adopted the DID regression to analyze the impact of the environmental pilot policy on productivity. They found that market-based environmental regulation could promote the productivity of industrial enterprises. Cai and Ye [6] took China’s new environmental protection law (NEPL) as the quasi-natural experiment to evaluate the influence of environmental regulation on enterprises TFP. They found that ecological policy NEPL effectively hindered the total factor productivity of enterprises, and this effect showed heterogeneity for different types of enterprises. Yang et al. [7] found that environmental regulations contribute to the improvement of R & D expenditure and thus to the development of productivity with industry-level panel data from 1997 to 2003. This result supports the Porter Hypothesis that stricter environmental regulations may strengthen instead of reducing industrial competitiveness. Hancevic [8] used data from 1985 to 1999 to explore the impact of the Clean Air Act Amendment on the productivity of coal-fired boilers. The results show that the productivity of the industry has declined by 1 to 2.5 percent due to the impact of environmental policies. Tang et al. [9] took China’s “Acid rain and sulfur dioxide pollution control areas” policy as the quasi-natural experiment and used the panel data of Chinese industrial enterprises from 1998 to 2007 to analyze the impact of environmental regulations on enterprises TFP by means of the DID estimation method. The results show that environmental regulations severely hindered the growth of enterprises TFP, and this kind of negative impact is lagging and persistent. Aldieri et al. [10] found that strict and positive environmental policies are conducive to the improvement of the technical efficiency of enterprises, and then increase the productivity of enterprises. 

The second is to study the impact of environmental regulation on regional air quality. Feng et al. [11] studied the impact of the “Air Pollution Control and Prevention Action Plan” on air quality according to data of major pollutants in 338 cities of China from 2013 to 2017. They found that the APCP Action Plan contributes to enhancing overall air quality and its impact on different regions is heterogeneous. Based on environmental regulations implemented in Beijing from 2008 to 2019, Han et al. [12] found that the air quality in Beijing has gradually improved in the past decade. Feng et al. [13] analyzed the spatial correlation of $PM_{2.5}$ in the Beijing–Tianjin–Hebei region of China with 2006–2018 data and identified different spatial correlation characteristics of $PM_{2.5}$ concentration in different urban groups, which are influenced not only by local environmental regulations, but also by the regulations implemented in neighboring cities. Li et al. [14] calculated the environmental TFP and its constituent elements by DEA, analyzed the influence of environmental efficiency on the concentration of $PM_{2.5}$ in China, and found that environmental efficiency has significant spatial spillover effect on $PM_{2.5}$.

The third is to study the impact of environmental regulation on regional environmental development. Wu et al. [15] analyzed the relationship between environmental regulation and green development by the spatial Dubin model and the panel model with Chinese provincial-level data from 2005 to 2016. The results display that there is a “U” shaped relationship between environmental control and green energy efficiency. Zarzoso et al. [16] used the environmental policy strictness index issued by the OECD as an indicator of environmental policy to analyze the impact of environmental regulations on TFP with the panel quantile regression model. They found that more stringent environmental regulations can promote cleaner production processes and energy efficiency. Lin and Chen [17] took China’s non-ferrous metal industry as a sample to discuss the relationship between environmental regulation and energy environmental performance, which is represented by the competitiveness index constructed with DEA in two stages. It concluded that there is a significant U-shaped nonlinear relationship between environmental regulation and EEPT.

Mazzanti et al. [18] used various methods to measure environmental productivity and conducted econometric analysis on long macroeconomic panel data sets to evaluate the impact of environmental policies. They found that environmental policies adopted by different countries, directly and indirectly, promoted the performance and environmental productivity at various environmental levels through innovation. Hille et al. [19] used policy and patent data from 194 countries and regions to study how different renewable energy support policies affect the impact of solar and wind energy technology innovation, and found that the combination of renewable energy support policies increases the number of patents for solar and wind energy-related technologies. Yang et al. [20] used the DID estimation method to study the relationship of the carbon intensity constraint policy proposed by the Chinese government in 2009 on green production performance. They found that there is a non-linear inverted U-shaped effect on the green productivity of Chinese industry, and after a brief ascent, industrial GPP showed a circuitous downward trend.

The fourth is to study the impact of environmental policy on green total factor productivity. For this aspect of research, scholars hold different views. Firstly, some scholars believe that environmental policy can promote the improvement of *GTFP*. Qiu et al. [21] used the DID model to study the impact of LCCP policies implemented in China in 2012 on urban *GTFP*, recognizing that LCCP policies do have a significant positive impact on the *GTFP* of pilot cities.

The promoting effect of policies on *GTFP* in different types of cities is heterogeneous. Based on provincial panel data from 2007 to 2017, Zhang et al. [22] used the DID estimation method to analyze the impact of the APCP policy implemented in 2013 on the *GTFP* of China’s chemical industry. They found that the policy significantly increased China’s chemical industry *GTFP* and technological progress is the major driving factor.

Some scholars have found that the impact of environmental policy on *GTFP* is nonlinear and heterogeneous. Based on the TFP data of 273 cities in China from 2003 to 2013, Li and Wu [23] calculated the ML index and adopted the spatial Dubin model to study the impact of environmental regulation on *GTFP*, and found that due to different political attributes of cities, the impacts of environmental regulation on *GTFP* in different regions are significantly different. Wang et al. [24] measured *GTFP* with the SBM-DDF method and analyzed the effect of environmental regulatory policies on *GTFP*, verifying the Porter Hypothesis that environmental policy has a positive impact on *GTFP* within a certain degree of strictness, but the impact becomes negative when the environmental policy exceeds a certain degree. Zhang and Vigne [25] studied the impact of financing-pollution mitigation policy instruments on corporate performance using the DID estimation methods based on data from manufacturing enterprises in Jiangsu Province. They found that financing and emission reduction policies have a penalty effect on the performance of high-polluting enterprises, including total factor productivity, profitability and sales growth.

Throughout the above research, scholars at home and abroad mainly focus on the impact of environmental regulation on regional productivity development, and provide a lot of relevant theoretical and empirical research methods on the relationship between environmental policy and regional sustainable development. However, there are still few studies on the sustainable development effect of the APCP Action Plan. Some scholars have analyzed the impact of *GTFP* on various industries, but little attention is given to its impact on the sustainable development of China’s provinces. The goal of a comprehensive environmental policy is not only to improve air quality and improve the environmental level, but also to promote regional sustainable development. Therefore, based on the existing research, this paper studies the impact and mechanism of the APCP Action Plan on the *GTFP* by using panel data from 30 provinces in China from 2004 to 2017. Compared with previous studies, the marginal contribution of this paper is mainly reflected in the following three aspects. Firstly, by analyzing the impact of the APCP Action Plan on the *GTFP* of Chinese provinces, this paper enriches the research theme in this field, rather than simply exploring the impact of environmental policies on productivity and air quality. In this paper, the SBM-DEA model is used to measure the *GTFP* of the provinces. The Panel OLS-DID model and the fixed effect model are used to explain that the APCP Action Plan can enhance the improvement of regional *GTFP*, which provides a useful reference for other countries to formulate environmental policies.

Moreover, since there are three pilot regions of the APCP Action Plan in China, and there are significant differences among the three pilot regions, this paper analyzes the heterogeneity of the impacts of the environmental policy on different pilot regions through sub-sample regression. Finally, by adding the rationalization of industrial structure and the optimization of industrial structure as moderating variables, this paper further studies whether there is a heterogeneous moderating effect in different moderating variables on the policy effect.

This paper is divided into five sections: Section 2 introduces the theory introduction and the research method explanation; Section 3 illustrates the empirical results and the analysis; Section 4 conducts a further discussion; the fifth section draws the conclusion. 

## 2. Materials and Methods

### 2.1. Theoretical Analysis and Research Hypothesis

The impact of environmental regulation on *GTFP* is uncertain. Environmental regulation affects the development of *GTFP* through the channels of the environmental cost, green technology innovation and so on. From the perspective of environmental cost brought by environmental regulation, with the intensifying of environmental regulation, the environmental cost faced by enterprises in the process of product production and processing will go up, and the profits of enterprises will decline gradually. It is difficult for enterprises to expand the production scale, which is not conducive to the expansion of reproduction, but also makes enterprises increase the amount of environmental pollution control [25]. From this point of view, environmental regulation will inhibit the improvement of *GTFP*. On the other hand, from the perspective of technological innovation, the strengthening of environmental regulation will promote enterprises to carry out green technological innovation [26], optimize and upgrade production processes to adapt to the high product quality standards [27], and reduce the waste of resources and environmental pollutants in the production process. Therefore, environmental regulation will promote the improvement of *GTFP*. As an important measure of environmental regulation, the environmental policy will also promote the development of *GTFP*. Specifically, it will promote the improvement of *GTFP* through technological innovation, resource allocation and industrial structure. The APCP Action Plan can enhance the upgrading of production technology, decrease pollution emissions, and significantly promote technological progress to drive *GTFP*. Local governments will strengthen the environmental supervision of enterprises according to the implementation of environmental policies, and enterprises need to adjust the industrial structure, product production, technological innovation to meet the requirements of the policy so as to improve the *GTFP*. Therefore, Hypothesis 1 is proposed as follows:

**Hypothesis** **1** **(H1).***The APCP Action Plan will promote the improvement of regional *GTFP**.

The intensity of environmental regulation is diverse in different regions. The regions with more pollutant emissions will be subject to greater environmental regulation intensity, thus enhancing more the local environmental level and improving the quality of people’s life. For industries with varying degrees of pollution, environmental regulation will have different levels of constraints and penalties, so the impact on the TFP of the industry is heterogeneous [28]. The imbalance of economic development level, industrial spatial structure and greening protection level among different regions is also one of the significant reasons for the regional heterogeneity of the influence of environmental regulation on *GTFP*. Similarly, the environmental policy will also have a heterogeneous impact on TFP at different enterprises, industries and regional levels [6], [29]. According to the difference of pollutant distribution, the environmental policy will also set up pilot areas and non-pilot areas, and the policy effect of pilot areas will be better than that of non-pilot areas [11]. Therefore, Hypothesis 2 is proposed as follows:

**Hypothesis** **2** **(H2).***The impact of the APCP Action Plan on *GTFP* in different regions has regional differences*.

There is a complex relationship among environmental regulation, industrial structure and *GTFP*. On the one hand, environmental regulation can promote the rational adjustment of industrial structure, reduce the proportion of high energy consumption and high pollution industries at first. Besides that, it can speed up green technological innovation, improve air quality and reduce pollutant emissions. These measures have a significant impact on *GTFP* [21]. On the other hand, the adjustment of industrial structure may not play a role in protecting the environment. With the development of a country’s economy, in the process of industrial transfer, the pollutant discharge of the secondary industry will be higher than that of the primary and tertiary industries, especially heavy industry, which will also consume a lot of natural resources and discharge more industrial wastewater, exhaust gas and solid waste to increase environmental pollution, reduce environmental quality and inhibit the increase of *GTFP*. It can be seen that the impact of industrial structure adjustment on *GTFP* has two sides, which need to be analyzed according to the focus of industrial structure transformation and optimization. Therefore, this paper proposes Hypotheses 3 and 4 as follows:

**Hypothesis** **3** **(H3).***The rationalization of industrial structure enhances the promotion effect of the APCP Action Plan on regional *GTFP**.

**Hypothesis** **4** **(H4).***The optimization of industrial structure weakens the promoting effect of the APCP Action Plan on regional *GTFP**.

### 2.2. Variables Description and Data Sources

#### 2.2.1. Variable Description 

In the process of regional sustainable economic development, the green total factor productivity (*GTFP*) is a concept with green background based on TFP and considering the undesirable output of pollutants. The *GTFP* is generally measured by input indicators such as labor, capital, and energy consumption, desired output indicators such as GDP, and undesired output indicators such as environmental pollution. *GTFP* emphasizes the coordinated development of economic growth and ecological environment, which can reflect the quality of economic growth. Therefore, it has become one of the core indicators for supervising regional sustainable development. The major explained variable in this paper is the provincial *GTFP*, which is calculated by the SBM-GML model. See Appendix A.1 for the detailed calculation process. Table 1 is a description of the variables used to calculate *GTFP*. According to the research and empirical analysis of other scholars [30,31,32], *GTFP* measured by the GML can used in the panel regression analysis.

The core explanatory variable of this paper is the dummy variable Policy [11]. On 13 September 2013, the State Council of China issued the APCP Action Plan, which presents specific requirements for regional remediation of fine particulate matter concentrations in the Beijing-Tianjin-Hebei Region in northern China, the Yangtze River Delta in central China and the Pearl River Delta region in southern China. Thus, the pilot regions of the APCP Action Plan policy are specific. Therefore, in this paper, the pilot provinces and cities (11 in total) (There are Beijing, Tianjin, Hebei, Shanxi, Shandong, Inner Mongolia Autonomous Region, Shanghai, Jiangsu, Zhejiang, Anhui and Guangdong provinces.) in the three target regions are selected as the treatment group affected by the policy. Other provinces (19 in total) are classified as the control group that did not receive significant policy effect. 

The moderating variables of this paper are rationalization of industrial structure (RIS) and optimization of industrial structure (OIS1, OIS2). The detailed calculation formula and process of moderating variables are given in Section A.2. The rationalization of industrial structure can effectively reflect the increasing degree of correlation and coordination between industries and the increasing level of effective utilization of resources, that is, to measure the coupling degree of factor input structure and output structure. In this paper, the Theil index is used to measure the rationalization of the industrial structure of each province. The index not only retains the advantage of the deviation of the industrial structure, but also measures the structural deviation between the output value of different industries and the employment. It also reflects the different economic status of each industry through the weighted output value. 

The optimization of industrial structure (OIS) in this paper includes the change of industrial structure proportional relationship (that is, the “quantity”, represented by OIS1) and the improvement of industrial structure degree (that is, the “quality”, represented by OIS2). The former refers to the shift of industrial focus, in turn, the transfer of various factors in turn, and the latter refers to the high added value, high technology and high intensification of industry. In this paper, the “quantity” of industrial structure optimization OIS1 is calculated by the industrial structure hierarchy coefficient, and the “quality” OIS2 is measured by the weighted value of the product of the output ratio of each industry and labor productivity.

To control the impact of other factors at the provincial level on *GTFP*, referring to scholars [8,20,22], this paper chooses the following controlling variables: urbanization level (UL), measured by the built-up area of each region, representing the impact of urbanization; industrialization level (IS), measured by the ratio of the output value of the secondary industry in each province to the regional GDP; R & D investment level (RD), measured by the ratio of R&D inputs to GDP in each province; population size (POP), measured by the resident population of each province at the end of the year; economic development level (EG), measured by the total industrial output value of each province; FDI level (FDI), measured by the actual use of FDI; cost-based environmental regulation (ER), measured by the total income of pollutant discharge in each province to measure.

The descriptive statistics of the whole sample variables are shown in Table 2. Descriptive statistics of the subsample areas are shown in Table A1, Table A2 and Table A3 in Appendix B. In order to eliminate the heteroscedasticity of the explained variable, the natural logarithm of the explained variable *GTFP* is taken.

#### 2.2.2. Data Sources

The sample of this paper includes 30 provinces in China from 2004 to 2017 (including municipalities and autonomous regions). Due to the lack of some variable data in Tibet, the data of Tibet are finally deleted. For the missing values of individual provinces, this paper uses the interpolation method to supplement them. Data of relevant variables used in this paper are derived from China Statistical Yearbook, China Urban Statistical Yearbook, China Land and Resources Statistical Yearbook, China Energy Statistics Yearbook, China Science and Technology Statistics Yearbook, China Environmental Statistics Yearbook, the EPS database and statistical yearbooks of 30 provinces.

### 2.3. Research Methods 

This paper focuses on the impact of environmental policy on *GTFP*, specifically testing the impact of the APCP Action Plan on regional *GTFP*, as well as the moderating effect with industrial structure optimization and rationalization as moderating variables.

Based on the research objectives, grouping dummy variable treated is set in this paper according to whether a province in China belongs to the pilot regions of the APCP Action Plan policy. The provinces within the three pilot regions specified by the policies are set as the treatment group and assigned a value of 1 (i.e., treated=1); provinces that are not within the policy pilot area are the control group and assigned a value of 0 (i.e., treated=0). Meanwhile, the dummy variable time is set according to the time of policy implementation. Considering the release date of the APCP Action Plan, which is 13 September 2013, and the data are annual panel data, the data of the year 2013 cannot reflect the effect of the policy, so this paper assigns 2014 and later years to 1 (i.e.,  time=1), and years prior to 2014 are assigned to 0 (i.e., time=0). By referring to the practice of Moser [33], this paper adds a time trend term to the regression model, i.e., $ytreat_{it}=treated_{it}*year$, which is the interaction item of the grouping dummy variable and time. This paper employs the Panel OLS-DID model and the fixed effect model to estimate changes in the differences estimation [5]. According to Hausman test, the fixed effects model is superior to the random-effects model, so the fixed effects model is selected for regression analysis. Formulas (1) and (2) are taken as benchmark models to test whether the impacts of the APCP Action Plan on the provincial *GTFP* are consistent with Hypotheses 1 and 2.
(1)$$lnGTFP_{it}=\alpha +\beta_{1}Policy_{it}+\gamma_{n}X_{int}+\beta_{2}ytreated_{i}+\tau_{t}+\mu_{i}+\varepsilon_{it}$$
(2)$$lnGTFP_{it}=\alpha +\beta_{1}Policy_{it}+\beta_{2}treated_{i}+\beta_{3}time_{t}+\gamma_{n}X_{int}+\varepsilon_{it}$$

The important premise of using the DID method is that the treatment group and the control group meet the parallel trend assumption [9]; that is, before the implementation of the APCP Action Plan policy, the *GTFP* of each province should remain relatively steady. Therefore, Model (3) is proposed to analyze whether this paper satisfies this basic assumption.
(3)$$lnGTFP_{it}=\alpha +\beta_{1}Policy_{it}+\beta_{2}treated_{i}+\beta_{3}time_{t}+\beta_{4}ytreat_{i}+\beta_{5}Before1+\beta_{6}Before2\;+\beta_{7}Current+\beta_{8}After1+\beta_{9}After2+\gamma_{1}X_{it}+\varepsilon_{it}$$

In exploring the moderating effect of industrial structure optimization and rationalization on the APCP Action Plan, we adopt the following moderating effect Models (4)–(6) with interaction terms:(4)$$lnGTFP_{it}=\alpha_{1}+\alpha_{2}TtR_{it}+\alpha_{3}Policy_{it}+\beta_{1}RIS_{it}+ytreat_{i}+\gamma_{n}X_{int}+\tau_{t}+\mu_{i}+\varepsilon_{it}$$
(5)$$lnGTFP_{it}=\alpha_{1}+\alpha_{2}TtO1_{it}+\alpha_{3}Policy_{it}+\beta_{1}OIS1_{it}+ytreat_{i}+\gamma_{n}X_{int}+\tau_{t}+\mu_{i}+\varepsilon_{it}$$
(6)$$lnGTFP_{it}=\alpha_{1}+\alpha_{2}TtO2_{it}+\alpha_{3}Policy_{it}+\beta_{1}OIS2_{it}+ytreat_{i}+\gamma_{n}X_{int}+\tau_{t}+\mu_{i}+\varepsilon_{it}$$

In the benchmark model, $G{\mathrm{TFP}}_{it}$ indicates the *GTFP* of province *i* in year *t*. In order to analyze the impact of the APCP Action Plan policy on provincial *GTFP*, this paper uses $Policy_{it}=treated*time$ as a dummy variable of policy influence. In the parallel trend analysis model, Before2, Before1, Current, After1, After2 indicates the policy effect variable of 2 years before, 1 year before, base year, 1 year after and 2 years after the implementation of the policy respectively. In the moderating effect models, three interactive variables $\;TtR_{it}=Policy_{it}*RIS$,$TtO1_{it}=Policy_{it}*OIS1$,$\;TtO2_{it}=Policy_{it}*OIS2$ represent the moderating effects of the industrial structure rationalization, the “quantity” and the “quality” of industrial structure optimization on the environmental policy. $X_{int}$ is the *n*-th control variable of province *i* in year *t*; $\tau_{t}$ represents the time effect; $\mu_{i}$ is the individual effect and $\varepsilon_{it}$ is the random interference term.

## 3. Empirical Results and Analysis

### 3.1. Econometric Test of the Benchmark Model

#### 3.1.1. Regression Results of the Benchmark DID Model

Based on the regression Models (1) and (2), this paper adopts the Panel OLS-DID estimation and the fixed effect model FE to carry on the regression analysis to the full sample, in order to study the impact of the APCP Action Plan policy on *GTFP* [34]. The regression results of the benchmark model are shown in Table 3.

The estimated regression coefficients for the variable Policy in Table 3 are significantly positive in Columns (1)–(4). For example, the 95% confidence interval for the coefficient in Column (1) is [0.0020,0.0274]. These results indicate that the implementation of the APCP Action Plan improved the average provincial *GTFP*.

The reason for this may be that the APCP Action Plan can effectively promote regional innovation and development of green technology, effectively limit the high pollution and high emission industries, promote the green transformation and optimization of the industrial structure, and reduce the concentration of air pollutants such as sulfur dioxide and $PM_{2.5}$ [11], thus increasing the *GTFP*. This also confirms Hypothesis 1. For this reason, in their efforts to achieve the goal of carbon neutrality and high-quality development, countries can reasonably use policy tools to promote green technological innovation [35] in high-pollution and high-emission industries. Moreover, it can reduce pollution emissions from this type of enterprise in the production process. Finally, it is important to improve energy efficiency [36], enhance overall air quality [12], and promote people’s happiness in life.

#### 3.1.2. Conditions for Application of the DID Model: Parallel Trend Test

The important premise of using the DID method is that the treatment group and the control group satisfy the parallel trend assumption; that is, before the implementation of the APCP Action Plan, the *GTFP* of each pilot province should maintain a relatively steady trend of change. In order to ensure this study satisfies this basic assumption, this paper adopts the pilot year 2014 as the base year. For the explained variable ln*GTFP* in the first two years and the next two years of the base year; the regression analysis with the Panel OLS-DID and the fixed effect (FE) model consistent with the benchmark regression is conducted separately. The regression results are shown in Table 4.

The coefficients of Before2 and Before1 are not significant in Table 4, indicating that there is no significant difference between the pilot and non-pilot provinces before 2014, which is in line with the parallel trend hypothesis. It shows that the DID method is suitable for the selection of benchmark model. Further, from the dynamic effect of the parallel trend test, the estimated parameters of the policy effect changed from negative coefficient before the environmental policy pilot to positive after the pilot. Therefore, it is distinct to reveal that the impact of the APCP Action Plan on *GTFP* is significantly positive. 

It is a significant that the change of the estimated parameter coefficient of the policy effect in Figure 1, which draws the conclusion that the assumption of the parallel trend is satisfied.

### 3.2. Robustness Test 

#### 3.2.1. The Placebo Test

In order to exclude the impact of other unknown factors on the selection of policy pilot provinces, it is essential to carry out a placebo test to ensure that the conclusion of this study is caused by the implementation of the APCP Action Plan policy. The Placebo test is a regression consistent with the benchmark regression by randomly selecting several virtual treatment groups in the whole sample, thus providing a robust guarantee for the conclusion of this study [8]. In this paper, 10,000 times of sampling are conducted in 30 provinces in China. In each sampling, 11 provinces were randomly selected as the virtual treatment group, and the remaining 19 provinces as the control group. Regression analysis was performed according to Panel OLS-DID and fixed-effect model.

In the kernel density distribution diagram of the core explanatory variable Policy, i.e., Figure 2 and Figure 3, the *x*-axis is the *t*-value of the policy coefficient of the core explanatory variable estimated from 11 provinces randomly selected for 10,000 times as the virtual treatment group, and the y-axis is the corresponding *p*-value. The curve represents the *t*-value distribution of the core explanatory variable policy coefficient. As can be seen from the above figure, the absolute values of the t value of most of the sampling estimation coefficients are less than 2, and the *p* values are above 0.1, which indicates that the APCP Action Plan has no significant effect in these 10,000 times random sampling. Therefore, the conclusion of this paper can be concluded as robust through the placebo test.

#### 3.2.2. Counterfactual Test

The premise of using the DID method is that the treatment group and the control group are comparable. If there is no influence of the APCP Action Plan policy, the *GTFP* of the treatment group and the control group will not change with time and produce significant differences. In this paper, the time of policy implementation is revised from 2014 to 2009 and 2010. According to benchmark Models (1) and (2), we conducted a test that was consistent with the benchmark regression. 

The estimated regression results of the counterfactual test are shown in Table 5. If the time of policy implementation is advanced to 2009 or 2010, the coefficients of Policy2009 and Policy2010 are not significant. The results of the counterfactual test show that before 2014, the APCP Action Plan policy has no significant impact on the *GTFP* of the treatment group and the control group; this means that the actual policy pilot year can significantly improve *GTFP*. Therefore, the conclusions of this paper are quite robust.

#### 3.2.3. Replacing the Explained Variable Test and the PSM-DID Estimation

In order to further verify the robustness of the conclusion, this paper analyzes whether the APCP Action Plan policy still has a promotional effect on *GTFP*1 by replacing the explained variable *GTFP* with *GTFP*1 calculated without carbon dioxide emissions by a regression analysis which is consistent with the benchmark Models (1) and (2).

Furthermore, the DID method is prone to the problem of “selection bias”. It is difficult to guarantee that the treatment group and the control group have the same individual characteristics before the implementation of the policy. The sample of this paper covers 30 provinces in the whole country, and there are great differences among the samples, such as in the aspects of territory, economic development, industrial structure and environmental situation, so there will be great individual differences in the sample data.

To solve the possible problems of the DID estimation, this paper matches the cities of the treatment group with those of the control group by using the propensity score matching method to take the control variable as the sample point. The matched data are further used for the DID regression, namely, the PSM-DID estimation [27]. By means of the PSM-DID estimation method, this paper further verifies the promoting effect of the APCP Action Plan policy on *GTFP*. The regression results of the above two methods are displayed in Table 6.

The estimated regression coefficients of the Policy in Table 6 are significantly positive in Columns (1)–(5). Columns (4) and (5) display the regression results estimated by the PSM-DID. This suggests that APCP Action Plan still significantly improves the *GTFP*. Thus, the conclusion of this paper is robust. 

To sum up, through a series of robustness tests, such as the placebo test, the counterfactual test, the explained variable replacement test and the PSM-DID estimation, this paper concludes that the promotion effect of the APCP Action Plan policy on *GTFP* is robust.

### 3.3. Heterogeneity Test 

In the process of the DID regression model, the econometric test and robustness test of the benchmark model indicate that the conclusion of this paper is significant and robust. In this paper, the APCP Action Plan significantly promotes the improvement of green total factor productivity. After the analysis of the above two parts, it is necessary to further explore the heterogeneity effect of the APCP Action Plan on *GTFP* in different pilot regions of China through the heterogeneity test, which will help the government to make further improvement plans for the implementation effect of policies in different regions, which is also the importance of the heterogeneity test. Based on Model (2), the Panel OLS-DID model is used to analyze the impact of the policy on *GTFP* in different regions. According to the principle of geographical proximity of Chinese provinces and the three key pilot areas specified in the APCP Action Plan, the full sample of China is divided into the Jing-Jin-Ji, Jin-Lu-Meng region in northern China (Beijing, Tianjin, Hebei, Shanxi, Shandong and Inner Mongolia autonomous region are in the treatment group, and Henan, Heilongjiang, Jilin, Liaoning, Shaanxi, Gansu, Ningxia, Xinjiang, Qinghai are in the control group;), the Yangtze River Delta region (with Shanghai, Zhejiang, Anhui, Jiangsu in the treatment group, and Jiangxi, Hunan, Hubei, Fujian, Sichuan and Chongqing in the control group;) in central China, the Pearl River Delta region (with Guangdong in the treatment group and Jiangxi, and Fujian, Hunan, Guangxi and Hainan are in the control group) in southern China. The empirical results of the regional heterogeneity test [37] are displayed in Table 7.

The estimated regression coefficients for the variable policy are significantly positive between the full sample and the southern China sample. However, the estimated coefficients of the variable policy between northern China and central China are not significant and negative, which indicates that the APCP Action Plan policy has heterogeneous effects on northern China, central China and southern China. The APCP Action Plan has contributed significantly to the *GTFP* improvement in southern China, while its impact on the *GTFP* of northern China and central China is not significant. 

Based on the empirical analysis of the benchmark model of subsamples in China, it is found that the impact of this policy on *GTFP* in different regions is heterogeneous. Through the empirical regression results in Table 7, it is clear that the policy has significantly enhanced the *GTFP* in southern China, and the other two regions have not been significantly affected by the policy. This may be due to the fact that the APCP Action Plan has different regulatory powers to different regions. There are differences in the degree of economic development, industrial structure, level of green technological innovation and environmental problems among different regions [38], so the impact of the policy on the *GTFP* in different regions has regional differences [6]. Different from the other two regions, the industrial structure of southern China is mainly secondary industry, and the air quality and environmental level are superior to the other two regions. The APCP Action Plan is conducive to the transformation and optimization of industrial structure and green technological innovation in southern China, thus promoting the development of a green economy in the region [39]. It also confirms Hypothesis 2.

As a consequence, when countries formulate corresponding environmental policies to solve pollution problems, they need to combine the emission of pollutants in different regions to achieve regional differentiated management and control. At the same time, regional joint prevention and control areas should be established to expand the scope of policy implementation [38], raise people’s vigilance against environmental pollution, and establish a new, greener environment.

## 4. Further Discussion

### 4.1. Mechanism Analysis

It is significant to indicate that the APCP Action Plan policy can improve *GTFP* in the above-mentioned research results. To further analyze the impact path, with reference to Wang [24] and Feng [27], this paper takes the rationalization of industrial structure RIS, the “quantity” of industrial structure optimization OIS1 and the “quality” of industrial structure optimization OIS2 as moderating variables. Regression of Models (4)–(6) was conducted to analyze the moderating effect of the APCP Action Plan on *GTFP* through the multi-dimensional transformation and optimization of industrial structure in the full sample area of China.

#### 4.1.1. Moderating Effect of Rationalization of Industrial Structure

Through analyzing the estimation coefficient of the interaction term TtR, this paper explores the moderating effect of the rationalization of the industrial structure RIS on the impact of the APCP Action Plan policy on *GTFP*.

The estimated coefficients for the interaction term TtR in Table 8 are significantly positive in Columns (1) and (2), indicating that the rationalization of industrial structure can significantly enhance the promotion effect of the APCP Action Plan on *GTFP*. Industrial structure rationalization is an important index that can effectively reflect the increasing degree of correlation and coordination between industries and the strengthening of effective utilization of resources [39]. After the promulgation of the APCP Action Plan, regions are subject to stricter environmental supervision [21]. Therefore, in order to improve the local environmental level and meet the minimum standards of environmental policy governance, various regions firstly carry out industrial structure transformation and optimization, make the industrial structure more rational. Secondly, improving the effective utilization of resources, and enhancing the coordination between industries to reduce pollutant emissions in the production process through green technology innovation and ultimately achieve sustainable development [40]. These also confirm Hypothesis 3.

#### 4.1.2. Moderating Effect of the Optimization of Industrial Structure

This paper analyzes the coefficient estimation of the interaction terms TtO1 and TtO2 between industrial structure optimization and policy so as to analyze the moderating effect of industrial structure optimization from the aspects of “quantity” and “quality”.

The regression results of the moderating effects for the “quantity” OIS1 and the “quality” OIS2 of the industrial structure optimization between Table 8 Columns (3) and (4) and Columns (5) and (6). The coefficient estimates of two interaction items TtO1 and TtO2 in Table 8 are significantly negative in Columns (3) and (4) and Columns (5)and (6), indicating that the high degree of “quantity” and “quality” of industrial structure optimization will weaken the promotion effect of the APCP Action Plan policy on *GTFP* [41]. Moreover, the “quantity” of industrial structure optimization weakens the promotion effect of environmental policy on *GTFP* more than the “quality “. From the comparison of the three interaction coefficients in Table 8, it can be seen that the moderating effect of the “quality” of industrial structure optimization is larger than that of industrial structure rationalization and that of the “quantity” of industrial structure optimization. The different emphasis of industrial structure has a heterogeneous moderating effect on the promotion effect of the APCP Action Plan on *GTFP*.

From the mechanism analysis, it can be seen that the moderating effect of the industrial structure has two sides. On the one hand, the rationalization of industrial structure can enhance the promotion of *GTFP* by the environmental policy. On the other hand, the high degree of “quantity” and “quality” of industrial structure optimization will weaken the promoting effect. At present, China is also carrying out the transformation and optimization of the industrial structure [42], shifting the focus of development from the primary industry to the second and third industries because the economic development is still dominated by heavy industry with high output value and high pollution. At present, the “quality” of the industrial structure optimization in China reflects the transition from the primary industry to the secondary industry. It is no doubt that the economic development of various regions largely depends on industry. Although this type of industrial transformation is an essential stage for a country’s economic prosperity [43], it is difficult to achieve the purpose of protecting and improving the environment. It usually aggravates industrial pollution and causes more resource consumption and pollutant discharge. The “quality” of the industrial structure optimization focuses on the transformation of industry to high output value and high technology. 

Under the supervision of environmental policy, various regions will reasonably renovate low output value and high pollution industries. At present, the industrial structure transformation of each region is still dominated by high output value and high pollution industries [44], and the high-tech and low-pollution tertiary industries are still in the development stage. It can be seen that the optimization of industrial structure, whether from the aspect of the “quality” or the “quantity“, will weaken the promotion of the environmental policy on *GTFP* at this stage. This is due to the characteristics of high yield and high pollution in the secondary industry [45], which confirms Hypothesis 4. Therefore, all regions should actively adjust the focus of industrial structure transformation and optimization according to actual local conditions at first.

Moreover, it is significant for them to strengthen environmental supervision of related enterprises with high pollution and high output value, adopt reasonable policies to promote local green technology innovation, and actively introduce high-tech enterprises, thus making the industrial structure transition from high-pollution industry to high-tech and high-output tertiary industry, and promoting the development of green economy in the region [46].

## 5. Conclusions

### 5.1. The Effect and Prospect of the APCP Action Plan

The APCP Action Plan requires that the concentrations of inhalable particulate matter in cities at or above the prefecture level in China will be reduced by more than 10% in 2017 compared with 2012. The three areas of focus are the concentration of fine particles ($PM_{2.5}$) in Jing-Jin-Ji, Yangtze River Delta and Pearl River Delta is about 25%, 20% and 15% lower than that in 2012. Therefore, the policy is called the most stringent air pollution control system in history. From the perspective of controlling pollutant emissions, the average concentration of inhalable particulate matter ($PM_{10}$) in cities of all regions and above in China decreased by 22.7% in 2017 compared with 2013, and that of Jing-Jin-Ji, Yangtze River Delta and Pearl River Delta decreased 39.6%, 34.3% and 27.7% compared with 2013. During the period 2013–2017, the average $PM_{2.5}$ concentration in China’s 30 provinces decreased from 52.71 μg/m^3^ to 39.94 μg/m^3^. The average sulfur dioxide emissions from industrial enterprises decreased from 681,100 tons to 217,600 tons. 

The average discharge of environmental pollutants, such as industrial wastewater, fixed waste generation and carbon dioxide, has been effectively controlled. From the perspective of the impact of human health, some research found that after APCP Action Plan, the loss of human health caused by air pollution has decreased, which avoids the premature death of more than 60,000 people. According to the survey results of the willingness to pay method, the increased health benefits are estimated to be 54.97 billion yuan. Therefore, the APCP action plan not only effectively improves the environmental level of each region, promotes the sustainable development of the local economy, but also alleviates people’s health crisis, and realizes the ecological mode of harmonious development between humans and nature.

Although the APCP Action Plan has gained remarkable achievements, the control of air pollution is a long process, and more long-term and effective policies are necessary. For example, the APCP Action Plan focuses on inhalable particulate matter and fine particulate matter, but there are no specific restrictions on the emission of volatile organic compounds and ozone. It also lacks the control of mobile source pollution. Therefore, the specific emission control indicators of other air pollutants should be further added to further promote the air quality standards. Based on this, a more effective and scientific policy mix [47] should focus on precise policy implementation, strengthening source control, scientific promotion and long-term mechanism.

Inspired by the APCP Action Plan, countries can consider the air remediation policies in key areas, adjust measures to local conditions step-by-step, and establish key prevention and control areas. When formulating environmental policies, various countries should combine sustainable development strategy and carbon-neutral concept at first. Secondly, it is necessary to make breakthroughs from the energy use structure and industrial structure of each region, reduce the utilization rate of fossil fuels, improve the utilization efficiency of new energy. Finally, different countries should enhance the level of scientific and technological innovation, which is conducive to the construction of innovative ecosystems, and realize the positive value of cocreation and mutual benefit.

### 5.2. Conclusion of the Paper

Based on the panel data of 30 provinces in China from 2004 to 2017, this paper analyzes the impact of the APCP Action Plan on *GTFP* [48]. Furthermore, it studies the moderating effects of industrial structure rationalization and optimization on the improvement of the environmental policy on *GTFP*. The following conclusions are drawn:

(1) The APCP Action Plan can effectively improve the *GTFP*, which indicates that environmental policies can improve the efficiency of resource utilization and reduce the emission of environmental pollutants and thus increase the *GTFP* by adjusting the industrial structure transformation and optimization among regions and improving the level of green technological innovation.

(2) The APCP Action Plan policy has heterogeneous effects on *GTFP* in different pilot regions. The policy has effectively promoted the increase of *GTFP* in the Pearl River Delta region in southern China, but its impact on the green development of the other two regions is not obvious. It shows that the APCP Action Plan policy has a regional difference in the impact on the *GTFP*.

(3) In the analysis of the mechanism, by introducing the industrial structure optimization and rationalization as the moderating variables, it is clear to find that the rationalization of the industrial structure can enhance the promotion effect of the APCP Action Plan policy on the *GTFP*.

In view of the conclusion, this paper puts forward the following relevant suggestions [49]. First, when formulating environmental policies, the government should consider regional differences, adjust measures to local conditions, and set different stage goals according to different pollution sources, rather than rigid policy provisions [50]. Second, the local governments can carry out the trade-off policy promotion work according to the environmental policy and the local development characteristics. On the one hand, the local governments should actively implement the relevant work of the policy and achieve the policy objectives by improving the transformation and optimization of industrial structure [51].

On the other hand, it is important for them to rationalize the transfer according to the local industrial characteristics without harming the local economic development as far as possible [52]. They should reduce the number of high-pollution and high-emission enterprises, improve the green technology innovation ability, reduce pollutant emissions, and then step into the road of sustainable development [53]. Third, it is essential for local governments to regulate and control environmental policies, promote the coordinated development of regional environment and economy, pay attention to the emphasis on industrial structure transformation and optimization. Moreover, speeding up the transition from primary industry to high output value and low pollution tertiary industry [54], and rationally adjusting the proportion structure of the secondary industry is necessary. All regions should actively eliminate enterprises with backward technology and serious pollution, invest more funds to encourage the inclusion of more high-tech enterprises, and promote the sustainable development of the economy [47,55,56].

## Figures and Tables

**Figure 1 ijerph-18-08216-f001:**
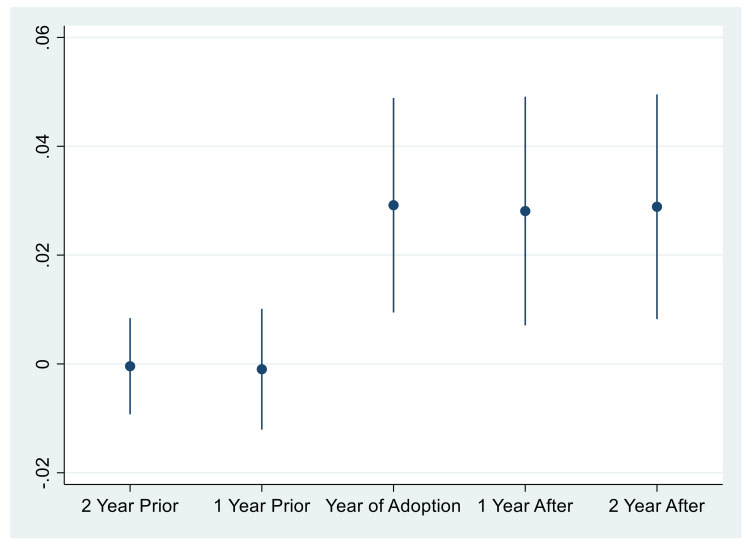
Box plot of parallel trend test (based on FE (4)).

**Figure 2 ijerph-18-08216-f002:**
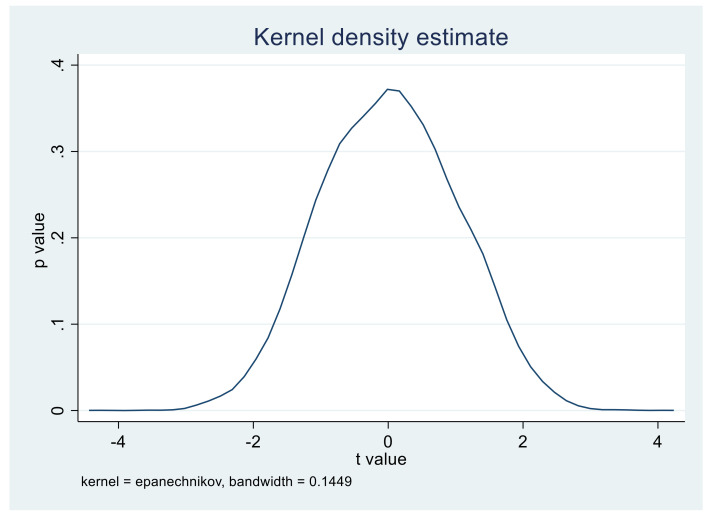
The panel OLS estimation graph of the policy estimation coefficient of 10,000 random sampling experiment panels with a bandwidth of 0.1449.

**Figure 3 ijerph-18-08216-f003:**
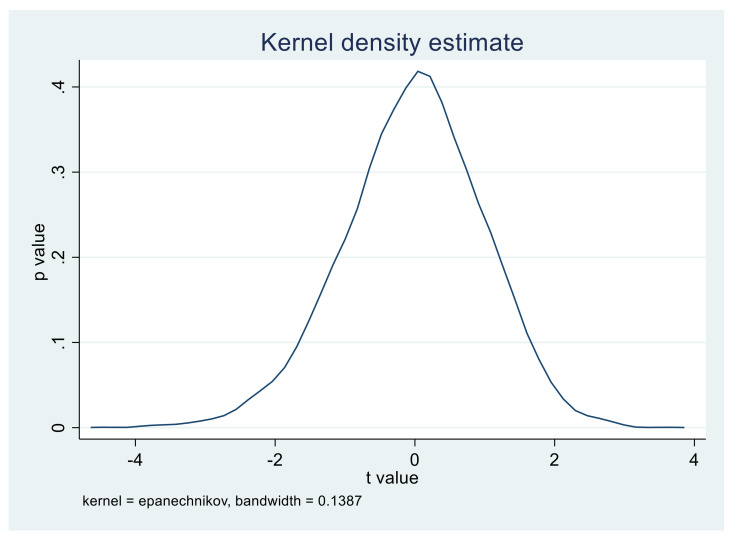
The graph of the policy estimation coefficient of 10,000 random sampling experiment panels with a bandwidth of 0.1387 (fixed-effect model).

**Table 1 ijerph-18-08216-t001:** Description of input–output indicators.

Input-Output Indicator		
Type	Indicator Name	Index Calculation Method
Input indicators	Capital	Perpetual Inventory method
	Labor	Number of employees in each province at the end of the period
	Energy consumption	Total energy consumption per province
Output indicators		
Desired output	Gross regional product	Provincial GDP for the year
Undesired output	Industrial wastewater discharge	Provincial industrial wastewater discharge
	Industrial fixed waste generation	Industrial fixed waste production in each province
	Industrial SO2 emissions	Provincial industrial sulfur dioxide emissions
	Carbon dioxide emissions	Provincial carbon dioxide emissions

**Table 2 ijerph-18-08216-t002:** Descriptive statistics of the full sample variables.

Variable	Obs	Mean	Std. Dev.	Min	Max
*lnGTFP*	420	−0.0492	0.055	−0.4332	0.0069
UL	420	1.4206	1.051	0.1033	5.8081
IS	420	0.4392	0.081	0.1690	0.6196
RD	420	0.0285	0.039	0.0002	0.2344
POP	420	0.4441	0.267	0.0539	1.1169
EG	420	0.5990	0.621	0.0098	3.5344
FDI	420	389.6732	455.917	0.0000	2300.0000
ER	420	5.7554	4.836	0.1563	28.7344

**Table 3 ijerph-18-08216-t003:** Parameter estimation results of the benchmark model.

	OLS (1)	FE (2)	OLS (3)	OLS (4)
	*lnGTFP*	*lnGTFP*	*lnGTFP*	*lnGTFP*
Policy	0.0151 **	0.0151 *	0.0152 *	0.0443 ***
	(2.44)	(1.96)	(1.74)	(3.89)
UL	0.0303	0.0303 ***	0.0215 ***	0.0183 ***
	(0.96)	(2.77)	(4.25)	(3.61)
RD	−1.9515 ***	−1.9515 ***	−1.6357 ***	−1.5614 ***
	(−7.30)	(−15.09)	(−7.26)	(−7.08)
EG	0.0836 **	0.0836 ***	0.0187	0.0259
	(2.12)	(6.26)	(1.17)	(1.61)
FDI	−0.00001	−0.00001	−0.00001	−0.00001
	(−1.00)	(−1.62)	(−0.35)	(−0.53)
POP	−0.0388	−0.0388	−0.0084	−0.0129
	(−0.34)	(−0.38)	(−0.75)	(−1.19)
ER	0.0006	0.0006	0.0013 **	0.0017 ***
	(0.90)	(0.91)	(2.33)	(3.11)
IS	−0.105 **	−0.105 **	−0.0129	−0.0330
	(−2.08)	(−2.41)	(−0.53)	(−1.42)
time			−0.0261 ***	−0.0287 ***
			(−6.40)	(−6.92)
treated			−0.0084 *	9.805 ***
			(−1.95)	(4.28)
ytreat	−0.0008	−0.0008		−0.0049 ***
	(−0.85)	(−0.84)		(−4.28)
Individual effect	YES	YES	YES	YES
Time effect	YES	YES	YES	YES
_cons	1.649	0.643	−0.0319 ***	−0.0225 **
	(0.83)	(0.89)	(−2.93)	(−2.16)
N	420	420	420	420

Note: *, ** and *** represent passing the significance test of 10%,5% and 1%, respectively, the same below.

**Table 4 ijerph-18-08216-t004:** OLS and FE estimates of parallel model trends.

	OLS (1)	OLS (2)	FE (3)	FE (4)
	*lnGTFP*	*lnGTFP*	*lnGTFP*	*lnGTFP*
Before2	−0.0064	0.0029	−0.0064	−0.0004
	(−1.10)	(0.51)	(−1.10)	(−0.10)
Before1	−0.0085	0.0030	−0.0083	−0.0010
	(−1.31)	(0.46)	(−1.28)	(−0.18)
Current	0.0208 **	0.0329 ***	0.0192 **	0.0292 ***
	(2.60)	(3.11)	(2.09)	(3.03)
After1	0.0188 **	0.0336 ***	0.0162	0.0281 **
	(2.38)	(3.29)	(1.65)	(2.73)
After2	0.0159 *	0.0323 ***	0.0167 *	0.0289 ***
	(1.97)	(3.55)	(1.88)	(2.86)
treated	−0.0082	6.752		
	(−0.93)	(1.51)		
time	−0.0272 ***	−0.0250 ***	−0.0270 ***	−0.0274 ***
	(−4.13)	(−4.23)	(−3.88)	(−3.86)
ytreat		−0.0034		−0.0033
		(−1.51)		(−1.50)
Control variables	YES	YES	YES	YES
Individual effects	YES	YES	YES	YES
Time effect	YES	YES	YES	YES
_cons	−0.0294	−0.0263	0.140	2.492
	(−1.15)	(−1.07)	(1.27)	(1.60)
N	420	420	420	420

Note: *, ** and *** represent passing the significance test of 10%,5% and 1%, respectively, the same below.

**Table 5 ijerph-18-08216-t005:** Benchmark model parameter estimates of the counterfactual test.

	OLS (1)	FE (2)	OLS (3)	FE (4)	OLS (5)	OLS (6)
	*lnGTFP*	*lnGTFP*	*lnGTFP*	*lnGTFP*	*lnGTFP*	*lnGTFP*
Policy2009		0.0012		0.0012	0.0027	
		(0.37)		(0.35)	(0.54)	
Policy2010	0.0058		0.0058			0.0062
	(1.64)		(1.64)			(1.11)
time					−0.0199 ***	−0.0178 ***
					(−8.67)	(−7.05)
treated					−0.0073 **	−0.0072 **
					(−2.16)	(−2.12)
Control variables	YES	YES	YES	YES	YES	YES
Individual effect	YES	YES	YES	YES	YES	YES
Time effect	YES	YES	YES	YES	YES	YES
_cons	−0.0264	−0.0293	0.0338	0.0275	−0.0184 **	−0.0223 ***
	(−1.01)	(−1.14)	(0.87)	(0.71)	(−2.42)	(−2.75)
N	300	300	300	300	300	300

Note: ** and *** represent passing the significance test of 5% and 1%, respectively, the same below.

**Table 6 ijerph-18-08216-t006:** Benchmark model and PSM-DID regression results.

	OLS (1)	FE (2)	OLS (3)	PSM-DID (4)	FE (5)
	*lnGTFP*1	*lnGTFP*1	*lnGTFP*1	*lnGTFP*	*lnGTFP*
Policy	0.0151 **	0.0151 **	0.0440 ***	0.0216 *	0.0124 ***
	(2.50)	(1.98)	(3.99)	(1.72)	(3.01)
time			−0.0285 ***	−0.0142 ***	
			(−6.93)	(−3.45)	
treated			10.05 ***	5.917 ***	
			(4.48)	(2.75)	
ytreat	−0.0010	−0.0010	−0.0050 ***	−0.0030 ***	
	(−0.97)	(−0.97)	(−4.48)	(−2.75)	
Control variable	YES	YES	YES	YES	YES
Individual effect	YES	YES	YES	YES	YES
Time effect	YES	YES	YES	YES	YES
_cons	1.872	0.715	−0.0301 ***	−0.0480 ***	0.00512
	(0.95)	(1.01)	(−2.87)	(−3.98)	(0.12)
N	420	420	420	354	354

Note: *, ** and *** represent passing the significance test of 10%,5% and 1%, respectively, the same below.

**Table 7 ijerph-18-08216-t007:** Regression results of the full sample and regional samples in China.

	Full Sample	Northern China	Central China	Southern China
	*lnGTFP*	*lnGTFP*	*lnGTFP*	*lnGTFP*
Policy	0.0152 *	0.0053	−0.0022	0.0114 *
	(1.74)	(0.52)	(−0.17)	(1.70)
treated	−0.0084 *	−0.0013	−0.0017	0.0187
	(−1.95)	(−0.25)	(−0.20)	(0.80)
time	−0.0261 ***	−0.0205 ***	−0.0306 ***	−0.0065 **
	(−6.40)	(−4.55)	(−3.00)	(−1.96)
Control variable	YES	YES	YES	YES
Individual effect	YES	YES	YES	YES
Time effect	YES	YES	YES	YES
_cons	−0.0319 ***	−0.0562 ***	−0.173 ***	−0.0547 ***
	(−2.93)	(−3.48)	(−3.30)	(−4.37)
N	420	210	140	84

Note: *, ** and *** represent passing the significance test of 10%,5% and 1%, respectively, the same below.

**Table 8 ijerph-18-08216-t008:** Test of the moderating effect of industrial structure on policy.

	OLS (1)	FE (2)	OLS (3)	FE (4)	OLS (5)	FE (6)
	*lnGTFP*	*lnGTFP*	*lnGTFP*	*lnGTFP*	*lnGTFP*	*lnGTFP*
TtR	0.161 **	0.178 ***				
	(2.12)	(4.13)				
TtO1			−0.256 ***	−0.266 ***		
			(−4.30)	(−3.62)		
TtO2					−0.0791 ***	−0.0803 ***
					(−3.96)	(−7.73)
Policy	−0.0168	−0.0104	0.627 ***	0.651 ***	0.0572 ***	0.0639 ***
	(−1.23)	(−1.06)	(4.40)	(3.68)	(5.26)	(6.61)
RIS	0.0133	0.0108				
	(0.65)	(0.44)				
OIS1			−0.0761 **	−0.0785 *		
			(−1.99)	(−1.82)		
OIS2					0.0057	0.0075
					(0.68)	(0.78)
ytreat		−0.0018 *		−0.00000141		−0.0012
		(−1.78)		(−0.34)		(−1.24)
Control variable	YES	YES	YES	YES	YES	YES
Individual effect	YES	YES	YES	YES	YES	YES
Time effect	YES	YES	YES	YES	YES	YES
_cons	−0.0192	1.351 *	0.184 *	0.216 **	−0.0223	0.907
	(−0.61)	(1.85)	(1.83)	(2.21)	(−0.80)	(1.34)
N	420	420	420	420	420	420

Note: *, ** and *** represent passing the significance test of 10%,5% and 1%, respectively, the same below.

## Data Availability

The data presented in this study are available on request from the author.

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
