# Peer review of "Does Environmental Policy Affect Green Total Factor Productivity? Quasi-Natural Experiment Based on China’s Air Pollution Control and Prevention Action Plan"

_ijerph, 2021, doi:10.3390/ijerph18158216_

Round 1

Reviewer 1 Report

First of all, I would like to thank the authors the opportunity of reading their work and raise my coments with the objective of improvimng the document quality.

I believe that the topic is very important however some ajustments have to be made to grant the publication. 

First of all, I suggest that the authors re-read the document and try to simplify the paragraphs, as most of them are too long. Also, I suggest an extensive proof reading, the text is hard to follow. Please try to cut the paragraphs and try to make the materials and methods less dense. The reader does not understant the underlying motivation for the different equations presented. 

Perhaps the reader would benefit from an explanation about the differences of the methods and the gains of each. All in all, some models appear as statistically insignificant, which in my view is of scant use, perhaps they should be removed. 

Then, putting more than 4 decimal places is unusual and reveals a model with very scant explanatory capacity. I think that it should be removed also. 

Moreover, the heterogeneity test should be included in a section which could be called robustness check.

An exhaustive description of the regions should be included as a footnote not as part of the text. 

Perhaps the models should be put together as it is very difficult to move back and foreward to compare the significances. 

Finally, in what concerns the conclusion section there should be more about policy recommendations. The debate should be extended to a proposal of policy mix. In this vein I suggest the addition of additional references in the topic: Costa, J. Carrots or Sticks: Which Policies Matter the Most in Sustainable Resource Management? Resources 202110, 12. https://doi.org/10.3390/resources10020012.

Please explain the state of the art of policy instruments and their effectiveness and suggest alternatives to raise the overall results, do not forget to connect with the SGDs and their expected impact on innovative ecosystems. 

I wish the authors the best of luck with their research.

Author Response

Dear Reviewer 1:

Thank you for your letter and for the Reviewer 1’s comments concerning our manuscript entitled “Does Environmental Policy Affect Green Total Factor Productivity? Quasi-Natural Experiment based on China's Air Pollution Control and Prevention Action Plan”. Those comments are all valuable and very helpful for revising and improving our paper, as well as the important guiding significance to our researches. We have studied comments carefully and have made corrections which we hope meet with approval. The revised portion is marked in red in the paper. The main corrections in the paper and the response to the Reviewer 1’s comments are as following:

Point 1: First of all, I suggest that the authors re-read the document and try to simplify the paragraphs, as most of them are too long. Also, I suggest an extensive proof reading, the text is hard to follow. Please try to cut the paragraphs and try to make the materials and methods less dense. The reader does not understant the underlying motivation for the different equations presented. 

Response 1: According to Reviewer1’s comments, we already simplified the paragraphs,materials and methods. Some statistical tables and variable descriptions have been included in Appendix A and Appendix B. We also deleted redundant models and empirical results.

Point 2: Perhaps the reader would benefit from an explanation about the differences of the methods and the gains of each. All in all, some models appear as statistically insignificant, which in my view is of scant use, perhaps they should be removed. 

Response 2: As Reviewer1’s suggested, we have removed redundant or insignificant models.

Point 3: Then, putting more than 4 decimal places is unusual and reveals a model with very scant explanatory capacity. I think that it should be removed also.

Response 3: According to Reviewer1’s comments, we have adjusted the units of the variables four decimal places after the decimal point to get the empirical results in Table 3 again. It is worth mentioning that changing the unit of these variables will not affect the significance of variables, nor will it change the significance and size of key explanatory variables, which is a data pre-processing process. This revision is in P.8, line 365.

Point 4: Moreover, the heterogeneity test should be included in a section which could be called robustness check.

Response 4: According to Reviewer1’s comments, we already adjusted the position of heterogeneity test. The reason for this adjustment is that in the DID method, after the preliminary conclusions of benchmark regression and robustness test are obtained, further test analysis is needed to illustrate the importance and heterogeneity of the conclusions in this paper. Therefore, we put the heterogeneity test after the benchmark regression and robustness test. This revision is in P.14, line 472 to P.16, line 518.

Point 5: An exhaustive description of the regions should be included as a footnote not as part of the text.

Response 5: As Reviewer1’s suggested, we have put a detailed description of the regions in the footnotes. The revisions are in P.6 and P.15.

Point 6: Perhaps the models should be put together as it is very difficult to move back and foreward to compare the significances.

Response 6: According to Reviewer1’s comments, in order to facilitate the comparison of models, we have combined the regulatory effect models of different regulatory variables. This revision is in P.16, line 533.

Point 7: Finally, in what concerns the conclusion section there should be more about policy recommendations. The debate should be extended to a proposal of policy mix. In this vein I suggest the addition of additional references in the topic: Costa, J. Carrots or Sticks: Which Policies Matter the Most in Sustainable Resource Management? Resources 202110, 12. https://doi.org/10.3390/resources10020012.

 Please explain the state of the art of policy instruments and their effectiveness and suggest alternatives to raise the overall results, do not forget to connect with the SGDs and their expected impact on innovative ecosystems

Response 7: As Reviewer1’s suggested, we already added the references firstly. Secondly, we have added a section 5.1, which can explain the state of the art of policy instruments and their effectiveness and suggest alternatives to raise the overall results. The explanations are in P.18, line 598 to P.18, line 635.

Reviewer 2 Report

The manuscript’ Does Environmental Policy Affect Green Total Factor Productivity? Quasi-Natural Experiment based on China's Air Pollution Control and Prevention Action Plan‘deals with  the studies the impact and mechanism of the APCP Action Plan on the GTFP by using  panel data from 30 provinces in China from 2004 to 2017. This manuscript is mainly reflected in the following three aspects: first, by analyzing the impact of the APCP Action Plan on the GTFP of Chinese provinces, this paper enriches the research theme in this field, rather than simply exploring the impact of environmental policies on productivity and air quality. The SBM-DEA model is used to measure the GTFP of the provinces, and the OLS-DID model  and the fixed effect model are used to explain that the APCP Action Plan can promote the  improvement of regional GTFP, which provides a useful reference for other countries to  formulate environmental policies.

Comments:

-Review figure 2 and 3, mainly the symbology used.

-Add in the discussion of results section how the hypotheses stated in the work were achieved or demonstrated.

Author Response

Dear Reviewer 2:

Thank you for your letter and for the Reviewer 2’s comments concerning our manuscript entitled “Does Environmental Policy Affect Green Total Factor Productivity? Quasi-Natural Experiment based on China's Air Pollution Control and Prevention Action Plan”. Those comments are all valuable and very helpful for revising and improving our paper, as well as the important guiding significance to our researches. We have studied comments carefully and have made corrections which we hope meet with approval. The revised portion is marked in red in the paper. The main corrections in the paper and the response to the Reviewer 2’s comments are as following:

Point 1: Review figure 2 and 3, mainly the symbology used.

Response 1: As Reviewer 2’s suggested, we already modified the symbols in figure 2 and 3. This revision is in P.12 line 416 to line 419.

Point 2: Add in the discussion of results section how the hypotheses stated in the work were achieved or demonstrated.

Response 2: According to Reviewer2’s comments, we have made further analysis and discussion on the results, and proved that the hypothesis is tenable. These explanations are in P.9, line 372 to line 376、P.15, line 503 to P.16, line 512,P.17, line 541 to line 548, P.17, line 575 to P.18, line 591.

Reviewer 3 Report

My comments are in the attached file.

Author Response

ear Reviewer 3:

Thank you for your letter and for the Reviewer 3’s comments concerning our manuscript entitled “Does Environmental Policy Affect Green Total Factor Productivity? Quasi-Natural Experiment based on China's Air Pollution Control and Prevention Action Plan”. Those comments are all valuable and very helpful for revising and improving our paper, as well as the important guiding significance to our researches. We have studied comments carefully and have made corrections which we hope meet with approval. The revised portion is marked in red in the paper. The main corrections in the paper and the response to the Reviewer 3’s comments are as following:

Point 1: Define the definition of GTFP in a manner that better relates to the context that the authors intend.

Response 1: According to Reviewer3’s comments, we have made a more appropriate definition of GTFP according to the focus of this paper. This explanation is in P.5, line 244 to line 251.

Point 2: Focus on the heterogeneity.

Response 2: As Reviewer 3’s suggested, we have put the heterogeneity test after the benchmark regression and robustness test, and explained the importance of heterogeneity test. This revision is in P.14, line 472 to P.16, line 518.

Point 3: Improve the placebo test.

Response 3: As Reviewer 3’s suggested, we already modified the symbols in figure 2 and 3.  This revision is in P.12 line 416 to line 419.

Point 4: Simplify the statistical presentation.

Response 4: According to Reviewer3’s comments, we have simplified the statistical presentation. The descriptive statistics by region and the statistical expression of empirical results have been adjusted. This revision is in P.21, Appendix B.

Point 5: Improve the communication of the material.

Response 5: As Reviewer 3’s suggested, in order to strengthen the connection of materials, we have put some variable calculation instructions and statistical descriptions of different regions in the appendix, and deleted some redundant parts. The revisions are in P.19 to P.21, Appendix A and Appendix B.

Point 6: Is the question original and well defined? Do the results provide an advance

in current knowledge?

Response 6: According to Reviewer3’s comments, this question has been defined. Based on the existing literature, we would further explore the impact of APCPAP on GTFP. This explanation is in P.4, line 164 to line 166.

Point 7: Are the conclusions interesting for the readership of the Journal? Will the

paper attract a wide readership, or be of interest only to a limited number of people?

The authors ask an important question: was the APCPAP effective as measured by the GFTP, which is an environmental-motivated revision of an historically dominant empirical concept? If they can adjust their measure of GFTP appropriately and give more attention to the findings of heterogeneity across the regions (which they treat only as “Further Discussion” rather than as an important result), then they will have results that merit publication in IJERPH rather than in journal that publishes brief accounts of empirical findings. A revised version of this paper and IJERPH would be a good match for each other as it would bring issues of economic development directly into interdisciplinary empirical assessments of environmental policy.

Response 7: According to Reviewer3’s comments, first of all, this article will be more suitable for publishing in Journal. Secondly, we have a suitable measure of GTFP, and put the heterogeneity test in a more critical position, adding some content to illustrate the importance of heterogeneity test. This revision is P.14, line 472 to P.16, line 518.

Point 8: Are the results interpreted appropriately? Are they significant? Are all conclusions

justified and supported by the results? Are hypotheses and speculations carefully identified as

such?

Response 8: According to Reviewer3’s comments, we have a reasonable explanation for the empirical results, and they are meaningful. The empirical conclusions and theoretical hypotheses are based on the empirical results of further analysis.

Round 2

Reviewer 1 Report

Thanks for considering the suggestions made. 

In my opinion the document has significantly improved. 

Best of luck with your research.

Author Response

Dear Reviewer 1:

Thanks for your letter concerning our manuscript entitled “Does Environmental Policy Affect Green Total Factor Productivity? Quasi-Natural Experiment based on China's Air Pollution Control and Prevention Action Plan”. Thank you very much for your advice and blessing, we will continue to revise the article.

Reviewer 3 Report

I summarized my five initial comments broadly as:

  1. Define the definition of GTFP in a manner that better relates to the context that the authors intend,
  2. Focus on the heterogeneity, and
  3. Improve the placebo test.
  4. Simplify the statistical presentation and
  5. Improve the communication of the material.

I am satisfied with the authors’ responses to (2) and to (4).

Working in reverse order and so starting with (5), I said:

       “As an example of improving the writing, note that the authors say, “The regression results in Table 7 show that the coefficients of Before 2 and Before 1 are not significant, and the regression coefficients are near 0.” (lines 464-465)  While the sentence is technically correct, the second clause is redundant.  This example is the type of editing error that a final pass through the manuscript that a native English speaker should catch that is not always obvious to even a skilled non-native speaker.

       I do have two comments about the References section.  First, not all of the references have the year of publication listed for them.  Second, because the review form asks referees about the extent of self-citations, I believe that only of the authors (T.L.) has papers listed in the References and each one seems to be appropriate.”

The authors replied: “Response 5: As Reviewer 3’s suggested, in order to strengthen the connection of materials, we have put some variable calculation instructions and statistical descriptions of different regions in the appendix and deleted some redundant parts.  The revisions are in P.19 to P.21, Appendix A and Appendix B.”

It does not appear that the authors conducted a final pass through the manuscript by a native speaker to resolve the errors in the use of English; notably, the sentence I used as an example remains exactly the same (see lines 393-394 in the revision).  Further, several of the references still lack the year of publication.

Given that the Journal is published in English, the writing within the manuscript needs to improve in many places.  As with the example I noted above, the writing is technically correct and readable; the manuscript simply does not read well.  Either the authors should go through the document carefully with a native speaker or one of the Journal’s copy editors will have to work with the authors to improve the manuscript.

With respect to (3), I said originally:

       “The Placebo Test the authors employ (Section 3.2.1 beginning on line 478) is a version of the resampling technique, but it is better than the more typical bootstrap confidence interval because it matches their context well.  I have four critiques of their test.

       First, the authors present a “kernel density distribution diagram of the explained variable lnGTFP” (line 493).  That is not the random variable of interest: the regression coefficient of the variable Policy (from Table 6, Column 3) is.

       Second, rather than a kernel density function, the authors should mimic the idea of a bootstrap confidence interval and present (say) a 95% interval for the regression coefficient of Policy.  This will give readers a simpler sense as to the likely ranges of the effectiveness of the APCPAP than the kernel density will give (because the readers won’t have to worry about the technical specifications of the density).

       Third, the authors should employ this placebo interval technique for their regional analyses and for their industrial structure analyses as well.  Here, I am both complimenting the authors for their choice of this analytical methodology and encouraging them to use this useful tool more often.

       Finally, I believe that using 1,000 random draws (line 484) for the aggregate analysis is an insufficient number given the regional heterogeneity that the authors found.  Since the number of combinations of choosing 11 treatment provinces from 30 equals 54,627,300, I would be more comfortable seeing the results of 10,000 random draws.  I assume that such a change is simply a matter of changing the software coding in a loop to increase the number of repetitions.”

The authors replied: “Response 3: As Reviewer 3’s suggested, we already modified the symbols in figure 2 and 3.  This revision is in P. 12 line 416 to line 419.”

This material now begins on line 406, and the statement “kernel density distribution diagram of the explained variable lnGTFP” is now in line 420.  The authors’ work on this topic and description of it in the paper remains unchanged.   So, my assessment remains unchanged.

With respect to (1), I said originally:

       “The authors focus on assessing the APCPAP.  According to the authors, the APCPAP was issued “[i]n view of the PM2.5 events in China in 2013” (line 57-58) and it “puts forward specific requirements for regional remediation of fine particulate matter concentrations” (lines 262-264).  The authors’ focus on particulate matter is consistent with other descriptions of the intentions behind the promulgation of the APCPAP (e.g., https://policy.asiapacificenergy.org/node/2875).  So, the measure of the GTFP must include measures that reflect particulate matter.  Yet, the authors (Table 5, line 410) include only SO2 and CO2 as direct air pollution measures.  Further, the inclusion “energy consumption” as a production input does not indirectly measure particulate matter as that variable does not differentiate between fuel types (e.g., shifting from coal to natural gas) or reflect measures that might reduce particulate matter from burning coal.  To meet their claims of accessing the APCPAP, the authors must revise their measure of GTFP to match its intentions.”

The authors replied: “Response 1: According to Reviewer 3’s comments, we have made a more appropriate definition of GTFP according to the focus of this paper.  This explanation is in P.5, line 244 to line 251.” 

The authors made only editorial changes and still do not include a measure of PM2.5 in their calculation of GTFP.  Although the authors claim that they are assessing the APCPAP, by ignoring PM2.5 they are undercutting their claims as reducing PM2.5 is the policy’s primary goal.  All they have to do is to add PM2.5 to their calculations of GTFP.  Alternatively, and this would require additional editing to shift the focus of the analysis to the policy’s indirect effects, they authors would have to develop plausible arguments that the changes in the air pollution variables they include –SO2 and CO2 – are indirectly related to PM2.5.

Author Response

Dear Reviewer 3:

Thanks for your letter concerning our manuscript entitled “Does Environmental Policy Affect Green Total Factor Productivity? Quasi-Natural Experiment based on China's Air Pollution Control and Prevention Action Plan”. The comments are all precious and very useful for revising and improving our paper, as well as the significant guiding important to our researches. We have studied suggestions carefully and have made corrections which we hope meet with approval. The main corrections in the paper and the response to your comments are as following:

Point 1: Working in reverse order and so starting with (5), I said:

     (1) “As an example of improving the writing, note that the authors say, “The regression results in Table 7 show that the coefficients of Before 2 and Before 1 are not significant, and the regression coefficients are near 0.” (lines 464-465)  While the sentence is technically correct, the second clause is redundant.  This example is the type of editing error that a final pass through the manuscript that a native English speaker should catch that is not always obvious to even a skilled non-native speaker. I do have two comments about the References section.  First, not all of the references have the year of publication listed for them.  Second, because the review form asks referees about the extent of self-citations, I believe that only of the authors (T.L.) has papers listed in the References and each one seems to be appropriate.”

The authors replied: “Response 5: As Reviewer 3’s suggested, in order to strengthen the connection of materials, we have put some variable calculation instructions and statistical descriptions of different regions in the appendix and deleted some redundant parts.  The revisions are in P.19 to P.21, Appendix A and Appendix B.”

It does not appear that the authors conducted a final pass through the manuscript by a native speaker to resolve the errors in the use of English; notably, the sentence I used as an example remains exactly the same (see lines 393-394 in the revision).  Further, several of the references still lack the year of publication.

Given that the Journal is published in English, the writing within the manuscript needs to improve in many places.  As with the example I noted above, the writing is technically correct and readable; the manuscript simply does not read well.  Either the authors should go through the document carefully with a native speaker or one of the Journal’s copy editors will have to work with the authors to improve the manuscript.

Response 1: By carefully reading your comments and suggestions, we have made corresponding changes in this article. First, we have added the year of publication to the references. The revisions are in P.21 line 741 to line 746, P.22 line 811 to line 812, line 817 to line818, P.23 line 854 to line 855.

Secondly, we have made a lot of amendments to the redundant part of this paper and the use of English, and invited native English speakers to make further amendments to the paper. These revisions are P.1 line 35 to line 40, P.4 line 156 to line 158, line 184 to line 188, line 191 to line 195, P,5 line 211 to line 216, P,7 line 299 to line 302, line 314 to line 316, P.8 line 328 to line 330, P.9 line 371 to line 375, P.10 line 397 to line 400, P.11 line 417 to P.13 line 429, line 435 to line 436, P.14 line 465 to line 467, line 469 to line 472, line 477 to line 486, P.15 line 493 to line 500, P.16 line 527 to line 529, P.17 line 536 to line 539, line 557 to line 560, P.18 line 619 to line 622, P.19 line 657 to line 659.

Point 2: With respect to (3), I said originally:

       “The Placebo Test the authors employ (Section 3.2.1 beginning on line 478) is a version of the resampling technique, but it is better than the more typical bootstrap confidence interval because it matches their context well.  I have four critiques of their test.

       First, the authors present a “kernel density distribution diagram of the explained variable lnGTFP” (line 493).  That is not the random variable of interest: the regression coefficient of the variable Policy (from Table 6, Column 3) is.

       Second, rather than a kernel density function, the authors should mimic the idea of a bootstrap confidence interval and present (say) a 95% interval for the regression coefficient of Policy.  This will give readers a simpler sense as to the likely ranges of the effectiveness of the APCPAP than the kernel density will give (because the readers won’t have to worry about the technical specifications of the density).

       Third, the authors should employ this placebo interval technique for their regional analyses and for their industrial structure analyses as well.  Here, I am both complimenting the authors for their choice of this analytical methodology and encouraging them to use this useful tool more often.

       Finally, I believe that using 1,000 random draws (line 484) for the aggregate analysis is an insufficient number given the regional heterogeneity that the authors found.  Since the number of combinations of choosing 11 treatment provinces from 30 equals 54,627,300, I would be more comfortable seeing the results of 10,000 random draws.  I assume that such a change is simply a matter of changing the software coding in a loop to increase the number of repetitions.”

The authors replied: “Response 3: As Reviewer 3’s suggested, we already modified the symbols in figure 2 and 3.  This revision is in P. 12 line 416 to line 419.”

This material now begins on line 406, and the statement “kernel density distribution diagram of the explained variable lnGTFP” is now in line 420.  The authors’ work on this topic and description of it in the paper remains unchanged.   So, my assessment remains unchanged.

Response 2: As Reviewer 3’s suggested, we have revised the paper accordingly. The first is line 423's variable correction of kernel density distribution in placebo test. We use 10000 random sampling tests to plot the estimated coefficient of policy into a kernel density distribution, as shown in Figure 2 and Figure 3. From Figure 2-3, we can clearly see that after 10000 random sampling tests, the range of T value of policy estimation coefficient is mainly between [-2,2], which shows that the conclusion of this paper is still robust. These revisions are in P.11, line 415 to P.13, line 427.

Secondly, we present a 95% interval for the region coefficient of policy after the benchmark regression empirical results in Table 3, which helps readers better understand the promotion effect of APCPAP on GTFP. The revision is in P.9, line 369 to line 373.

Finally, we have made a further explanation about “Third, the authors should employ this placebo interval technique for their regional analyses and for their industrial structure analyses as well.”Placebo test in DID method is designed for the robustness of the empirical results of benchmark regression. In DID method, parallel trend test and placebo test are usually needed to show the robustness of benchmark regression, and then further regional heterogeneity analysis and mechanism analysis are used to expand the scope of this study. At the same time, we also refer to the overall framework of some scholars for this type of article [1-3], and then design benchmark regression, placebo test, heterogeneity analysis, industrial structure analysis and so on. Therefore, we thank you very much for your valuable suggestions, we still think that this paper may not need to add placebo test in heterogeneity analysis and industrial structure analysis.

  1. Zhang, Y., Song, Y., & Zou, H. Transformation of pollution control and green development: Evidence from China's chemical industry. J Environ Manage, 275,111246. doi:10.1016/j.jenvman.2020.111246
  2. Qiu, S., Wang, Z., & Liu, S. The policy outcomes of low-carbon city construction on urban green development: Evidence from a quasi-natural experiment conducted in China. Sustainable Cities and Society, 66. doi:10.1016/j.scs.2020.102699
  3. Zhang, D., & Vigne, S. A. The causal effect on firm performance of China's financing-pollution emission reduction policy: Firm-level evidence. Journal of Environmental Management, 279. doi:10.1016/j.jenvman.2020.111609

    Point 3: With respect to (1), I said originally:

           “The authors focus on assessing the APCPAP.  According to the authors, the APCPAP was issued “[i]n view of the PM2.5 events in China in 2013” (line 57-58) and it “puts forward specific requirements for regional remediation of fine particulate matter concentrations” (lines 262-264).  The authors’ focus on particulate matter is consistent with other descriptions of the intentions behind the promulgation of the APCPAP (e.g., https://policy.asiapacificenergy.org/node/2875).  So, the measure of the GTFP must include measures that reflect particulate matter.  Yet, the authors (Table 5, line 410) include only SO2 and CO2 as direct air pollution measures.  Further, the inclusion “energy consumption” as a production input does not indirectly measure particulate matter as that variable does not differentiate between fuel types (e.g., shifting from coal to natural gas) or reflect measures that might reduce particulate matter from burning coal.  To meet their claims of accessing the APCPAP, the authors must revise their measure of GTFP to match its intentions.”

    The authors replied: “Response 1: According to Reviewer 3’s comments, we have made a more appropriate definition of GTFP according to the focus of this paper.  This explanation is in P.5, line 244 to line 251.” 

    The authors made only editorial changes and still do not include a measure of PM2.5 in their calculation of GTFP.  Although the authors claim that they are assessing the APCPAP, by ignoring PM2.5 they are undercutting their claims as reducing PM2.5 is the policy’s primary goal.  All they have to do is to add PM2.5 to their calculations of GTFP.  Alternatively, and this would require additional editing to shift the focus of the analysis to the policy’s indirect effects, they authors would have to develop plausible arguments that the changes in the air pollution variables they include –SO2 and CO2 – are indirectly related to PM2.5.

    Response 3: According to Reviewer3’s comments, we give the following reasons to show that GTFP does not need to make additional changes.

    First of all, starting from the research focus of this paper, the GTFP measured in this paper is to study how APCPAP further affects the green sustainable development of provinces by affecting the industrial structure, rather than the air quality analysis of inter provincial regions. We hope to study the impact of the ten environmental protection policies on regional sustainable development from the perspective of green development. It is so significant that paying more attention to the economic effects of the policies on the region, not just the role of environmental protection.

    Secondly, SO2 is directly related to PM2.5. Studies have shown that there is a positive correlation between SO2 and PM2.5, sulfur oxide is one of the important components of PM2.5, and it is a gaseous pollutant with large quantity and wide influence range in the current air pollutants [1-3]. Because SO2 and NOx emitted from coal combustion of thermal power plant will produce nitrate, sulfate and other substances through chemical reaction, resulting in the increase of PM2.5 concentration. Therefore, the reduction of SO2 emission, that is, the implementation of high-intensity SO2 control measures, is also helpful to reduce PM2.5 emission. There is a certain impact relationship between SO2 and PM2.5. Previous studies believe that PM2.5 particulate pollution mainly comes from automobile exhaust and agricultural straw combustion, and these two activities also produce a lot of CO2. From the perspective of correlation between SO2, CO2 and PM2.5 and environmental pollutants, the GTFP calculated in this paper is in line with the theme and can reflect the green sustainable development level of a province.

    Finally, we would like to thank reviewer 3 for providing so many valuable and useful opinions and comments. The follow-up research of this paper can be carried out in the direction of PM2.5.

    1. Zhang, Y., Liu, X., Zhang, L., Tang, A., Goulding, K., & Collett, J. L. J., Jr. Evolution of secondary inorganic aerosols amidst improving PM2.5 air quality in the North China plain. Environmental Pollution, 281. doi:10.1016/j.envpol.2021.117027
    2. Liu, X., Wang, M., Pan, X., Wang, X., Yue, X., Zhang, D., . . . He, H. Chemical formation and source apportionment of PM2.5 at an urban site at the southern foot of the Taihang mountains. Journal of Environmental Sciences, 103, 20-32. doi:10.1016/j.jes.2020.10.004
    3. Zhao, C., Sun, Y., Zhong, Y., Xu, S., Liang, Y., Liu, S., . . . He, M. Spatio-temporal analysis of urban air pollutants throughout China during 2014-2019. Air Quality Atmosphere and Health. doi:10.1007/s11869-021-01043-5

Round 3

Reviewer 3 Report

Referring to the numbering in my initial report, my points (1), (3), and (5) remained when I wrote my second report.

  1. Let me quote from lines 50-52 of the paper: "In view of the PM2.5 events in China in 2013, the Chinese government issued the 'Air Pollution Control and Prevention Action Plan' (APCP Action Plan for short)." And two lines later, the authors say: "Based on this, A quasi-natural experiment based on the APCP Action Plan can help fully reflect the relationship between environmental policies and green total factor productivity."  In the conclusion (lines 601-603), the authors say: "The end of 2017, APCP Action Plan requires that the concentrations of inhalable particulate matter in cities at or above the prefecture level in China will be reduced by more than 10% in 2017 compared with 2012."

Given what the authors say, a reader who is relatively uniformed about Chinese environmental policy must naturally expect the analysis to focus on PM2.5.  Even if the authors dropped the clause "In view of the PM2.5 events in China in 2013" as well as the material in the conclusion about particulate matter, a reader who is somewhat informed about Chinese environmental policy would know the linkage between the APCP Action Plan and PM2.5.  So, quite frankly, I can't "accept" a manuscript that has this specific motivation and focus but does not include PM pollution data (see Table 1, line 256).

I would accept this paper for publication if the authors followed my original suggestion of revising their dependent variable to include PM.  Alternatively, the authors would have to do a serious revision of their introduction and conclusion (and likely their literature review as well).  The point of this revision would be to make that argument that a policy intended to affect one pollution could affect firms' control responses and outcomes for other pollutants as well.  In their case, the authors would have to discuss how controlling PM would likely affect wastewater discharge, fixed waste generation, SO2 emissions, and CO2 emissions (see Table 1, line 256).  Making this argument well is possible, but it will require that the authors develop it carefully.

  1. In line 415 the authors refer to “1,0000” and in line 426 the authors refer to “10000”, so it appears that the authors did increase their random draws from 1,000 to 10,000.

However, the authors’ use of English in the writing of what Figures 2 and 3 represent makes it difficult to understand exactly what they did to generate those figures.  In my original report I said: “First, the authors present a ‘kernel density distribution diagram of the explained variable lnGTFP’ (line 493).  That is not the random variable of interest: the regression coefficient of the variable Policy (from Table 6, Column 3) is.”  What the authors currently say in lines 423-426 is:

            “In the kernel density distribution diagram of the core explanatory variable Policy [notice how they are again referring to the variable not to the coefficient], i.e., Figure 2 and Figure 3, the absolute values of the T value of most of the sampling estimation coefficients [now the authors appear to be referring to the sampling distribution of the t-statistic associated with the coefficient of Policy] are less than 2, and the P values are above 0.1 [neither Figure 2 nor Figure 3 graphs p-values], which indicates that the APCP Action Plan has no significant effect in these 10000 times random sampling.”

So, the authors may well be presenting appropriate information, but given that they do not explain what they did well it is hard to be certain.  The burden is on the authors to communicate clearly.

  1. Again, I need to reiterate my comment that the authors need have a native speaker read through and edit the paper. Some of the quotes that I provided about contain examples of writing that such editing would identify and improve.  Again, in the spirit that some aspects of the writing are technically correct but awkward, here are two other examples.
    1. In line 182, the authors say: “The impact of environmental regulation on GTFP is uncertain but mainly positive”. At this point in the paper, the authors are still asking the question, what is the impact of regulation on GTFP?  They should not be answering it.  So, the authors should say: “The impact of environmental regulation on GTFP is uncertain but mainly positive”.
    2. In lines 368-373, the authors say: “Table 3 illustrates the regression results of the impact of the APCP Action Plan on GTFP under the full sample of China. According to the regression results in Table 3, for all Chinese provinces, the estimated results of the coefficient Policy are significantly positive in Columns (1) - (4) of Table 3, indicating that the APCP Action Plan promote the improvement of GTFP.  The 95% interval for the OLS-DID (1) regression coefficient of Policy is [0.0020,0.0274].”  The authors might say: “The estimated regression coefficients for the variable Policy in Table 3 are significantly positive in Columns (1) – (4).  For example, the 95% confidence interval for the coefficient in Column (1) is [0.0020,0.0274].  These results indicate that the implementation of the APCP Action Plan improved the average provincial GTFP.”

Author Response

Dear Reviewer 3:

Thanks for your letter concerning our manuscript entitled “Does Environmental Policy Affect Green Total Factor Productivity? Quasi-Natural Experiment based on China's Air Pollution Control and Prevention Action Plan”. The comments are all precious and important for revising and improving our paper, as well as the significant guiding important to our researches. We have studied suggestions carefully and have made corrections which we hope meet with approval. The revised portion is marked in red in the paper. The main corrections in the paper and the response to your comments are as following:

Point 1: Let me quote from lines 50-52 of the paper: "In view of the PM2.5 events in China in 2013, the Chinese government issued the 'Air Pollution Control and Prevention Action Plan' (APCP Action Plan for short)." And two lines later, the authors say: "Based on this, A quasi-natural experiment based on the APCP Action Plan can help fully reflect the relationship between environmental policies and green total factor productivity."  In the conclusion (lines 601-603), the authors say: "The end of 2017, APCP Action Plan requires that the concentrations of inhalable particulate matter in cities at or above the prefecture level in China will be reduced by more than 10% in 2017 compared with 2012."

Given what the authors say, a reader who is relatively uniformed about Chinese environmental policy must naturally expect the analysis to focus on PM2.5.  Even if the authors dropped the clause "In view of the PM2.5 events in China in 2013" as well as the material in the conclusion about particulate matter, a reader who is somewhat informed about Chinese environmental policy would know the linkage between the APCP Action Plan and PM2.5.  So, quite frankly, I can't "accept" a manuscript that has this specific motivation and focus but does not include PM pollution data (see Table 1, line 256).

I would accept this paper for publication if the authors followed my original suggestion of revising their dependent variable to include PM.  Alternatively, the authors would have to do a serious revision of their introduction and conclusion (and likely their literature review as well).  The point of this revision would be to make that argument that a policy intended to affect one pollution could affect firms' control responses and outcomes for other pollutants as well.  In their case, the authors would have to discuss how controlling PM would likely affect wastewater discharge, fixed waste generation, SO2 emissions, and CO2 emissions (see Table 1, line 256).  Making this argument well is possible, but it will require that the authors develop it carefully.

Response 1: Thanks for your suggestions. It is our pleasure to receive your valuable comments. We also quite agree with your opinions. As you said, the research focus of APCP Action Plan is PM2.5. The policy is closely related to PM2.5. Every reader familiar with China's policy will certainly be looking forward to the effects of PM2.5 in this article. Through this environmental policy, this paper studies whether the economic development of each province will be damaged under the premise of ensuring the gradual improvement of the environment. When considering the undesired output of GTFP, since SO2 is highly correlated with PM2.5, and PM2.5 emission reduction measures will also affect the emission of other pollutants, PM2.5 is not included in the calculation of GTFP in this paper. However, after the suggestions and analysis of reviewer 3, we believe that this is a very important research direction in the future. We are very grateful to reviewer 3 for broadening our research ideas. According to Reviewer3’s comments, we decided to revise the introduction and conclusion of this article to discuss how controlling PM would likely affect wastewater discharge, fixed waste generation, SO2 emissions, and CO2 emissions. There are many measures in the APCP Action Plan to effectively control the above environmental pollutants, which are also included in the calculation of GTFP in this paper. We hope to focus on the impact relationship between APCP Action Plan and GTFP in this paper, and further study the relationship between APCP Action Plan and PM2.5 in the future. The revisions are in P.2 line 46 to line 82, P.4 line 179 to line 184, P.18 line 630 to line 635, line 639 to line 649.

Point 2: In line 415 the authors refer to “1,0000” and in line 426 the authors refer to “10000”, so it appears that the authors did increase their random draws from 1,000 to 10,000.

However, the authors’ use of English in the writing of what Figures 2 and 3 represent makes it difficult to understand exactly what they did to generate those figures.  In my original report I said: “First, the authors present a ‘kernel density distribution diagram of the explained variable lnGTFP’ (line 493).  That is not the random variable of interest: the regression coefficient of the variable Policy (from Table 6, Column 3) is.”  What the authors currently say in lines 423-426 is:“In the kernel density distribution diagram of the core explanatory variable Policy [notice how they are again referring to the variable not to the coefficient], i.e., Figure 2 and Figure 3, the absolute values of the T value of most of the sampling estimation coefficients [now the authors appear to be referring to the sampling distribution of the t-statistic associated with the coefficient of Policy] are less than 2, and the P values are above 0.1 [neither Figure 2 nor Figure 3 graphs p-values], which indicates that the APCP Action Plan has no significant effect in these 10000 times random sampling.”

So, the authors may well be presenting appropriate information, but given that they do not explain what they did well it is hard to be certain.  The burden is on the authors to communicate clearly.

Response 2: Thanks for your suggestions. Because of your careful review, the article can become more excellent and understandable. As Reviewer 3’s suggested, we have revised the information of Figure 2 and Figure 3 and added corresponding explanations to make it easier to understand the effect of the placebo test in this article. The revisions are P.12 line 442 to P.13 line 456.

Point 3: Again, I need to reiterate my comment that the authors need have a native speaker read through and edit the paper. Some of the quotes that I provided about contain examples of writing that such editing would identify and improve.  Again, in the spirit that some aspects of the writing are technically correct but awkward, here are two other examples.

  1. In line 182, the authors say: “The impact of environmental regulation on GTFP is uncertain but mainly positive”. At this point in the paper, the authors are still asking the question, what is the impact of regulation on GTFP?  They should not be answering it.  So, the authors should say: “The impact of environmental regulation on GTFP is uncertain but mainly positive”.
  2. In lines 368-373, the authors say: “Table 3 illustrates the regression results of the impact of the APCP Action Plan on GTFP under the full sample of China. According to the regression results in Table 3, for all Chinese provinces, the estimated results of the coefficient Policy are significantly positive in Columns (1) - (4) of Table 3, indicating that the APCP Action Plan promote the improvement of GTFP.  The 95% interval for the OLS-DID (1) regression coefficient of Policy is [0.0020,0.0274].”  The authors might say: “The estimated regression coefficients for the variable Policy in Table 3 are significantly positive in Columns (1) – (4).  For example, the 95% confidence interval for the coefficient in Column (1) is [0.0020,0.0274].  These results indicate that the implementation of the APCP Action Plan improved the average provincial GTFP.”

Response 3: Thanks for your suggestions and examples. The examples are very reasonable and make us realize that there are still many shortcomings in this article. According to Reviewer3’s comments, we have revised some aspects of the writing are technically correct but awkward in the paper. We also have invited native speaker to make detailed modifications to this paper. All the modifications in this paper are marked in red.
